# Supertoroidal light pulses as electromagnetic skyrmions propagating in free space

Yijie Shen [1✉], Yaonan Hou[1], Nikitas Papasimakis [1] & Nikolay I. Zheludev[1,2]

Topological complex transient electromagnetic fields give access to nontrivial light-matter interactions and provide additional degrees of freedom for information transfer. An important example of such electromagnetic excitations are space-time non-separable single-cycle pulses of toroidal topology, the exact solutions of Maxwell's equations described by Hellwarth and Nouchi in 1996 and recently observed experimentally. Here we introduce an extended family of electromagnetic excitation, the supertoroidal electromagnetic pulses, in which the Hellwarth-Nouchi pulse is just the simplest member. The supertoroidal pulses exhibit skyrmionic structure of the electromagnetic fields, multiple singularities in the Poynting vector maps and fractal-like distributions of energy backflow. They are of interest for transient light-matter interactions, ultrafast optics, spectroscopy, and toroidal electrodynamics.

[1] Optoelectronics Research Centre, Centre for Photonic Metamaterials, University of Southampton, Southampton SO17 1BJ, UK. [2] Centre for Disruptive Photonic Technologies, School of Physical and Mathematical Sciences and The Photonics Institute, Nanyang Technological University, Singapore 637378, Singapore. ✉email: y.shen@soton.ac.uk

Topology of complex electromagnetic fields is attracting growing interest of the photonics and electromagnetics communities[1–5], while topologically structured light fields find applications in super-resolution microscopy[6], metrology[7,8], and beyond[9,10]. For example, the vortex beam with twisted phase, akin to a Mobius strips in phase domain, can carry orbital angular momentum with tunable topological charges enabling advanced applications of optical tweezers, machining, and communications[9–12]. The complex electromagnetic topological strips, knots, and caustic structures were also proposed as novel information carriers[13–16]. Recently, skyrmions, as topologically protected quasiparticles in high-energy physics and magnetic materials[17], were also studied in optical electromagnetic fields. Optical skyrmionic fields were first demonstrated in the evanescent field of a plasmonic surface[18,19], followed by the spin field of confined free-space waves[20,21], Stokes vectors of paraxial vector beams[22–25], polarizations in momentum-space[26], and pseudospins in photonic crystals[27]. The sophisticated vector topology of optical skyrmions holds potential for applications in ultrafast nanometric metrology[28], deeply subwavelength microscopy[20], and topological Hall devices[27].

While a large body of work on topological properties of structured continuous light beams may be found in literatures, works on the topology of the time-dependent electromagnetic excitations and pulses only start to appear. For instance, the "Flying Doughnut" pulses, or toroidal light pulses (TLPs) first described in 1996 by Hellwarth and Nouchi[29], with unique spatiotemporal topology predicted recently[30], have only very recently observed experimentally[31]. Fueled by a combination of advances in ultrafast lasers and metamaterials in our ability to control the spatiotemporal structure of light[32,33] together with the introduction of experimental and theoretical pulse characterization methods[34–37], TLPs are attracting growing attention. Indeed, TLPs exhibit their complex topological structure with vector singularities and interact with matter through coupling to toroidal and anapole localized modes[38–41]. However, while higher order, supertoroidal modes in matter have been introduced in the form of the fractal iterations of solenoidal currents[42–47], generalizations of free-space propagating toroidal pulses have not been considered to date.

In this paper, we report that the Hellwarth and Nouchi pulses, are, in fact, the simplest example of an extended family of pulses that we will call supertoroidal light pulses (STLPs). We will show that supertoroidal light pulses introduced here exhibit complex topological structures that can be controlled by a single numerical parameter. The STLP display skyrmion-like arrangements of the transient electromagnetic fields organized in a fractal-like, self-affine manner, while the Poynting vector of the pulses feature singularities linked to the multiple energy backflow zones.

## Results

**Supertoroidal electromagnetic pulses.** Following Ziolkowski, localized finite-energy pulses can be obtained as superpositions of "electromagnetic directed-energy pulse trains"[48]. A special case of the localized finite-energy pulses was investigated by Hellwarth and Nouchi[29], who found the closed-form expression describing a single-cycle finite energy electromagnetic excitation with toroidal topology obtained from a scalar generating function $f(\mathbf{r}, t)$ that satisfies the wave equation $(\nabla^2 - \frac{1}{c^2}\frac{\partial^2}{\partial t^2})f(\mathbf{r}, t) = 0$, where $\mathbf{r} = (r, \theta, z)$ are cylindrical coordinates, $t$ is time, $c = 1/\sqrt{\varepsilon_0\mu_0}$ is the speed of light, and the $\varepsilon_0$ and $\mu_0$ are the permittivity and permeability of medium. Then, the exact solution of $f(\mathbf{r}, t)$ can be given by the modified power spectrum method[29,48], as $f(\mathbf{r}, t) = f_0/[(q_1 + i\tau)(s + q_2)^\alpha]$, where $f_0$ is a normalizing constant, $s = r^2/(q_1 + i\tau) - i\sigma$, $\tau = z - ct$, $\sigma = z + ct$, $q_1$ and $q_2$ are

parameters with dimensions of length and act as effective wavelength and Rayleigh range under the paraxial limit, while $\alpha$ is a real dimensionless parameter that must satisfy $\alpha \geq 1$ to ensure finite energy solutions. In particular, the parameter $\alpha$ is related to the energy confinement of the pulse with $\alpha < 1$ resulting in pulses of infinite energy, such as planar waves and cylindrical waves, while $\alpha \geq 1$ leads to finite-energy pulses. Next, transverse electric (TE) and transverse magnetic (TM) solutions are readily obtained by using Hertz potentials. The electromagnetic fields for the TE solution can be derived by the potential $\mathbf{A}(\mathbf{r}, t) = \mu_0\nabla\times\hat{\mathbf{z}}f(\mathbf{r}, t)$ as $\mathbf{E}(\mathbf{r}, t) = -\mu_0\frac{\partial}{\partial t}\nabla\times\mathbf{A}$ and $\mathbf{H}(\mathbf{r}, t) = \nabla\times(\nabla\times\mathbf{A})$[29,48]. Finally assuming $\alpha = 1$, the electromagnetic fields of the TLP are described by[29]:

$$E_\theta = -4if_0\sqrt{\frac{\mu_0}{\varepsilon_0}}\frac{r(q_1 + q_2 - 2ict)}{\left[r^2 + (q_1 + i\tau)(q_2 - i\sigma)\right]^3} \tag{1}$$

$$H_r = 4if_0\frac{r(q_2 - q_1 - 2iz)}{\left[r^2 + (q_1 + i\tau)(q_2 - i\sigma)\right]^3} \tag{2}$$

$$H_z = -4f_0\frac{r^2 - (q_1 + i\tau)(q_2 - i\sigma)}{\left[r^2 + (q_1 + i\tau)(q_2 - i\sigma)\right]^3} \tag{3}$$

where $E_\theta$ represents the azimuthally directed component of electric field, and $H_r$ and $H_z$ are the radially and longitudinally directed components of magnetic field, respectively. Note that the TE mode field does not possess other kinds of components except the three, and the TM mode is expressed by exchanging the electric and magnetic fields. Equations (1)–(3) derived by Hellwarth and Nouchi for $\alpha = 1$ show the simplest example of TLPs. Here we explore the general solution for values of $\alpha \geq 1$. In the TE case, electric and magnetic fields are given by (see detailed derivation in Supplementary Notes 1–3):

$$E_\theta^{(\alpha)} = -2\alpha if_0\sqrt{\frac{\mu_0}{\varepsilon_0}}\left\{\frac{(\alpha+1)r(q_1+i\tau)^{\alpha-1}(q_1+q_2-2ict)}{\left[r^2+(q_1+i\tau)(q_2-i\sigma)\right]^{\alpha+2}} - \frac{(\alpha-1)r(q_1+i\tau)^{\alpha-2}}{\left[r^2+(q_1+i\tau)(q_2-i\sigma)\right]^{\alpha+1}}\right\} \tag{4}$$

$$H_r^{(\alpha)} = 2\alpha if_0\left\{\frac{(\alpha+1)r(q_1+i\tau)^{\alpha-1}(q_2-q_1-2iz)}{\left[r^2+(q_1+i\tau)(q_2-i\sigma)\right]^{\alpha+2}} - \frac{(\alpha-1)r(q_1+i\tau)^{\alpha-2}}{\left[r^2+(q_1+i\tau)(q_2-i\sigma)\right]^{\alpha+1}}\right\} \tag{5}$$

$$H_z^{(\alpha)} = -4\alpha f_0\left\{\frac{(q_1+i\tau)^{\alpha-1}\left[r^2-\alpha(q_1+i\tau)(q_2-i\sigma)\right]}{\left[r^2+(q_1+i\tau)(q_2-i\sigma)\right]^{\alpha+2}} + \frac{(\alpha-1)(q_1+i\tau)^{\alpha-2}(q_2-i\sigma)}{\left[r^2+(q_1+i\tau)(q_2-i\sigma)\right]^{\alpha+1}}\right\} \tag{6}$$

For $\alpha = 1$, the electromagnetic fields in Eqs. (4)–(6) are reduced to that of the fundamental TLP Eqs. (1)–(3). Moreover, the real and imaginary parts of Eqs. (4)–(6), simultaneity fulfill Maxwell equations and therefore represent real electromagnetic pulses.

While propagating in free space, toroidal and supertoroidal pulses exhibit self-focusing. Figure 1 shows the evolution of the fundamental ($\alpha = 1$) TLP and STLP ($\alpha = 5$) upon propagation through the focal point. In the former case, the pulse is single-cycle at focus ($z = 0$) becoming $1\frac{1}{2}$-cycle at the boundaries of Rayleigh range $z = \pm q_2/2$ (Fig. 1a). On the other hand, STLPs with $\alpha > 1$ (Fig. 1b, $\alpha = 5$) exhibit a substantially more complex spatiotemporal evolution where the pulse is being reshaped multiple times upon propagation. The STLP also possesses a more complex singular vector field configuration than the fundamental TLP. For instance, the magnetic vector distribution (see insets to Fig. 1a, b) exhibits vortex-type singularities (gray lines) and saddle points (full circles), resulting in skyrmions structures in the transverse plane (colored arrows). See Supplementary Videos 1 and 2 for dynamic evolutions upon propagation of the fundamental TLP and STLP, and

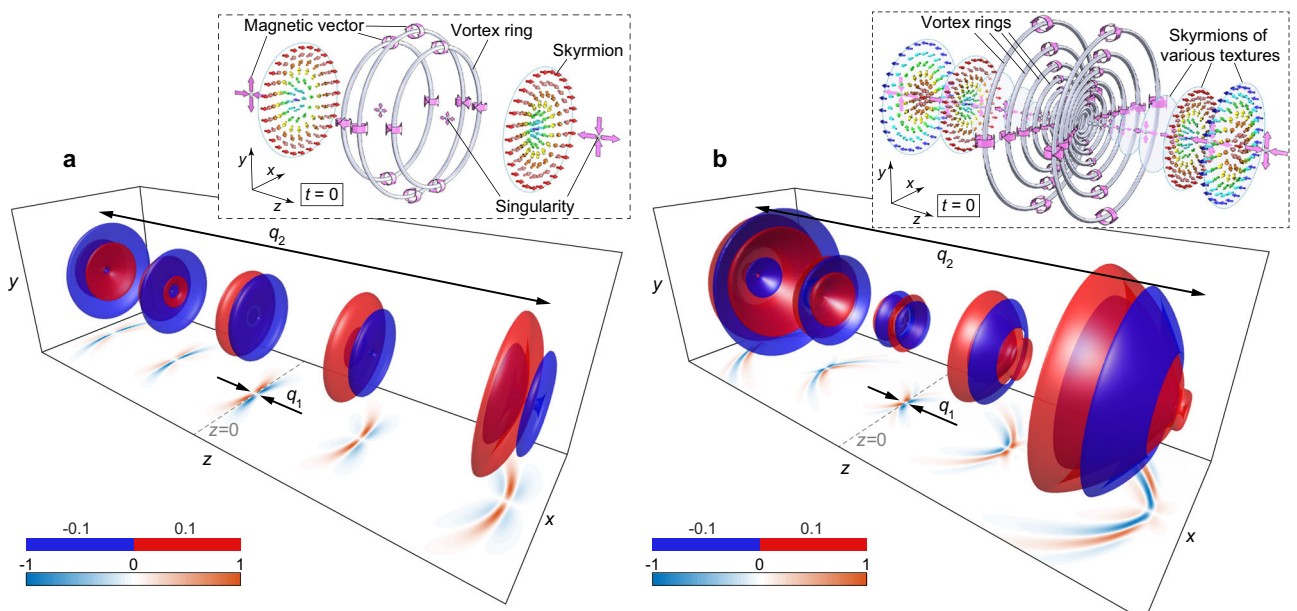

**Fig. 1 From toroidal to supertoroidal light pulses. a, b** Isosurfaces for the electric fields of **a** the fundamental TLP $Re[E_\theta(\mathbf{r}, t)]$, and **b** a STLP $Re[E_\theta^{(\alpha)}(\mathbf{r}, t)]$ of $\alpha = 5$, at amplitude levels of $E = \pm 0.1$ and the Rayleigh range of $q_2 = 100q_1$, at different times of $t = 0$, $\pm q_2/(4c)$, and $\pm q_2/(2c)$. $x$–$z$ cross-sections of the instantaneous electric field at $y = 0$. The insets in (**a**) and (**b**) are schematics of spatial topological structures of magnetic vector fields at focus ($t = 0$) for the fundamental TLP and STLP, respectively. The gray dots and rings mark the distribution of singularities (saddle points and vortex rings) in magnetic field, large pink arrows mark selective magnetic vector directions, and the smaller colored arrows show the skyrmionic structures in magnetic field.

Supplementary Video 3 for the evolution of TLP and STLP with different focused degrees versus values of $q_2/q_1$.

**Electric field singularities.** Figure 2 comparatively shows the instantaneous electric fields for the TE single-cycle fundamental TLP and STLP ($\alpha = 5$) with $q_2 = 20q_1$ at the focus ($t = 0$). In all cases, there are always central singularities on $z$-axis ($r = 0$; see vertical solid black lines in Fig. 2a, b) owing to the azimuthal polarization. For the fundamental TLP, the electric field possesses two shell-like singular surfaces symmetrically distributed along axis. The singular shells divide the parts with opposite chiralities of azimuthal polarization, as shown in Fig. 2a. Thus, the electric field forms counter-clockwise vortices around the $z$ axis at $z = q_1$ and $z = 35q_1$ (Fig. 2a1, a3). On the other hand, at $z = 5q_1$ (Fig. 2a2) the electric field vanishes on a singular shell, while it changes its orientation from clockwise close to the $z$-axis to counter-clockwise away from the axis. For the STLP case, a more complex matryoshka-like structure emerges with multiple nested singularity shells, replacing the single shell of the fundamental TLP, as Fig. 2b shows. The electric field configuration close to the singularity shells can be examined in detail at transverse planes at $z = q_1, 5q_1, 35q_1$ (Fig. 2b1–b3). In this case, at transverse planes close to $z = 0$, the electric field changes orientation from counter-clockwise close to $r = 0$ to clockwise away from the $z$-axis (see Fig. 2b1). On the other hand, on transverse planes close to $z = 5q_1$ (see Fig. 2b2), two singular rings (corresponding to the two singular shells) emerge as a cross-section of the multi-layer singularity shell structure, separating space in three different regions, in which the electric field direction alternates between counter-clockwise ($r/q_1 < 7$), to clockwise ($7 < r/q_1 < 15$), and again to counter-clockwise ($r/q_1 > 15$).

In general, the pulse of higher order of $\alpha$ is accompanied by a more complex multi-layer singular-shell structure, see the dynamic evolution versus the order index in Supplementary Video 4. Although the above results of electric fields are instantaneous at $t = 0$, we note that the multi-layer shall structure propagation of supertoroidal light pulse is retained during propagation, see such dynamic process in Supplementary Video 5.

**Magnetic field singularities.** The magnetic field of STLPs has both radial and longitudinal components, $\mathbf{H} = H_r\hat{\mathbf{r}} + H_z\hat{\mathbf{z}}$, which lead to a topological structure more complex than the one exhibited by the electric field. Figure 3 comparatively shows the instantaneous magnetic fields for the TLP and the STLP of $\alpha = 5$. For the fundamental TLP (Fig. 3a), the magnetic field has ten different vector singularities on the $x$–$z$ plane, including four saddle points on $z$-axis and six vortex rings (the surrounding vector distribution forming a vortex loop) away from the $z$-axis. We note that we only consider the singularities existed at an area containing 99.9% of the energy of the pulse. While the singularity existed at the region far away from the pulse center with nearly zero energy can be neglected. The magnetic field distribution in the vicinity of the singularities is shown in more detail in Fig. 3a1. Due to the axial symmetry of the pulse, the three singularities away from the $z$-axis correspond to three rings with vortices rotating clockwise ($z = 0$) or counter-clockwse ($z > 0$ and $z < 0$). For instance, Fig. 3a2 presents the magnetic field distribution in the $z = 0$ plane, where the orientation of the magnetic field changes from parallel to the $z$-axis within the vortex ring to anti-parallel outside the singularity ring. For the STLP (Fig. 3b), more vector singularities are unveiled in the magnetic field with six saddle points on $z$-axis and six off-axis singularities. The singularity structure of the STLP is presented in Fig. 3b1. The orientation of the magnetic field around the on-axis saddle points is alternating between "longitudinal-toward radial-outward point" and "radial-toward longitudinal-outward", similarly to the on-axis singularities of the TLP. Moreover, the off-axis singularities at $z = 0$ become now saddle points contributing to the singularity ring in the $z = 0$ plane. The remaining off-axis singularities are accompanied by clockwise and counter-clockwise magnetic field configurations at $x > 0$ and $x < 0$, respectively as shown Fig. 3b2.

**Skyrmionic structure in magnetic field.** An intriguing feature of the topology of STLPs involves the presence of skyrmionic quasiparticle configurations, which can be observed in the magnetic field topology of STLPs. The skyrmion is a topologically protected

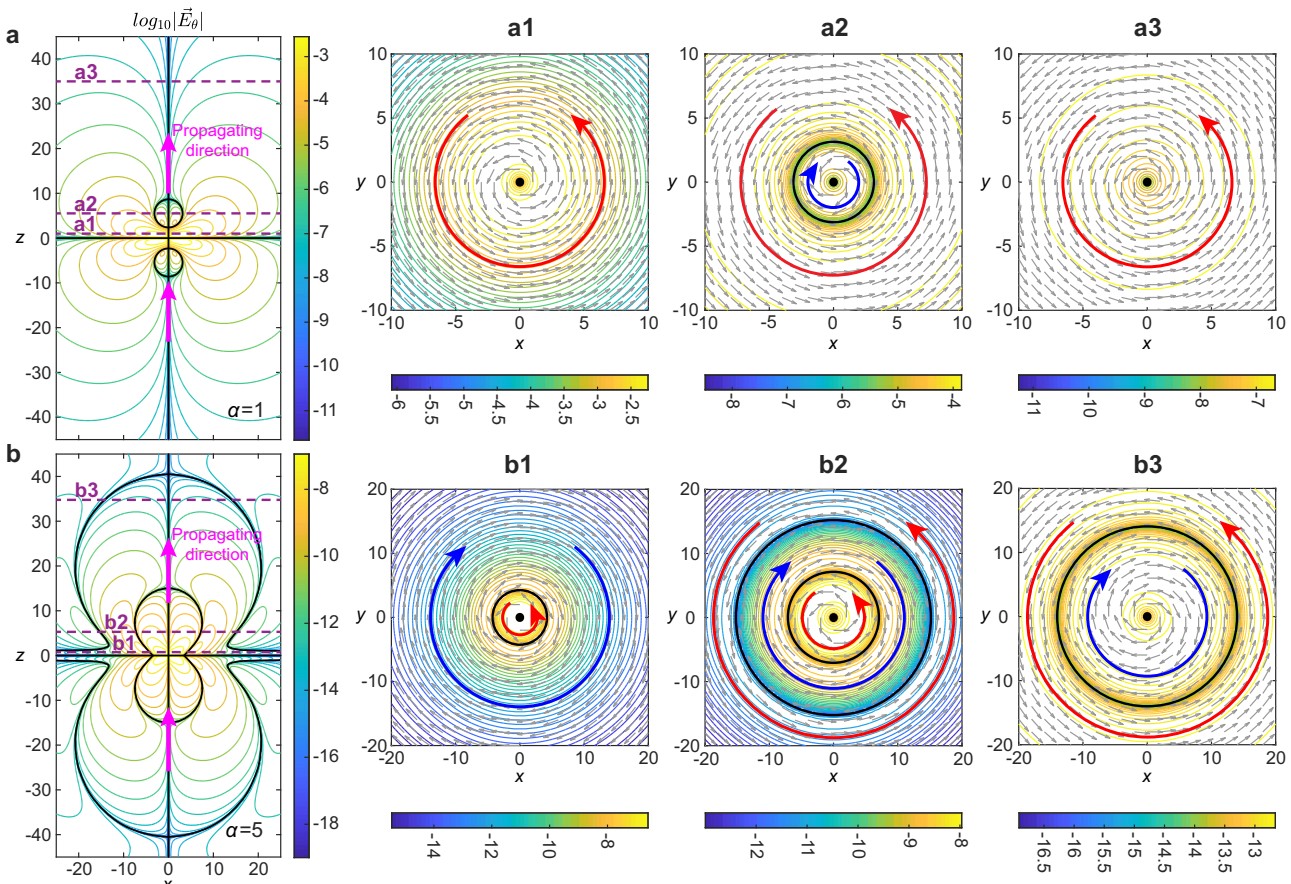

**Fig. 2 Electric fields of toroidal and supertoroidal light pulses. a, b** The isoline plots of the electric field in the $x$–$z$ plane for **a** the fundamental TLP, $Re[E_\theta(\mathbf{r}, t = 0)]$, and **b** the STLP of $\alpha = 5$, $Re[E_\theta^{(\alpha=5)}(\mathbf{r}, t = 0)]$, in logarithmic scale. The bold black lines represent the zero-value singular lines. The dashed purple lines represent the positions of propagation distance corresponding to the transverse plots at right side. Panels **a1–a3** and **b1–b3** show the electric field distributions in the transverse planes. The field magnitude is plotted as contours (in logarithmic scale), while the field orientation is presented by arrow plots. Electric field zeros are marked by the black solid bold lines and black dots. Blue and red arrows represent the two opposite azimuthal directions of the electric fields. Unit for coordinates: $q_1$.

quasiparticle with a hedgehog-like vectorial field, that gradually changes orientation as one moves away from the skyrmion center[49–51]. Recently skyrmion-like configurations have been reported in electromagnetism, including skyrmion modes in surface plasmon polaritons[18] and the spin field of focused beams[20,22]. Here we observe the skyrmion field configurations in the electromagnetic field of propagating STLPs.

The topological properties of a skyrmionic configuration can be characterized by the skyrmion number $s$, which can be separated into a polarity $p$ and vorticity number $m$[51]. The polarity represents the direction of the vector field, down (up) at $r = 0$ and up (down) at $r \to \infty$ for $p = 1$ ($p = -1$), the vorticity controls the distribution of the transverse field components, and another initial phase $\gamma$ should be added for determining the helical vector distribution, see "Methods" for details. For the $m = 1$ skyrmion, the cases of $\gamma = 0$ and $\gamma = \pi$ are classified as Néel-type, and the cases of $\gamma = \pm\pi/2$ are classified as Bloch-type. The case for $m = -1$ is classified as anti-skyrmion.

Here the vector forming skyrmionic structure is defined by the normalized magnetic field $\mathbf{H}/|\mathbf{H}|$ of the STLP. Two examples of two skyrmionic structures in the fundamental TLP are shown in Fig. 3c1 ($p = m = 1$, $\gamma = \pi$) and 3c2 ($p = m = 1$, $\gamma = 0$) occurre at the two transverse planes marked by purple dashed lines c1 and c2, which are both Néel-type skyrmionic structures, where the vector changes its direction from "down" at the center to "up" away from the center. In the case of the STLPs with more

complex topology, it is possible to observe more skyrmionic structures. The STLP pulse ($\alpha = 5$) exhibits not only the clockwise ($p = m = 1$, $\gamma = \pi$) and counter-clockwise ($p = m = 1$, $\gamma = 0$) Néel-type skyrmionic structures (Fig. 3c3, c4), but also those with $p = -1$, $m = 1$, $\gamma = \pi$ and $p = -1$, $m = 1$, $\gamma = 0$, in Fig. 3c5, c6.

In general, as the value of $\alpha$ increases, toroidal pulses show an increasingly complex magnetic field pattern with skyrmionic structures of multiple types, see Supplementary Video 4. We also note that the topology of the STLP is maintained during propagation, see Supplementary Video 5.

**Energy backflow and Poynting vector singularities.** The topological features of electromagnetic fields in supertoroidal pulses are linked to anomalous behavior of energy flow as represented by the Poynting vector $\mathbf{S} = \mathbf{E} \times \mathbf{H}$. An interesting effect for the fundamental TLP is the presence of energy backflow: the Poynting vector at certain regions is oriented against the pro-rogation direction (blue arrows in Fig. 4a)[30]. Such energy backflow effects have been predicted and discussed in the context of singular superpositions of waves[52,53], superoscillatory light fields[7,54], and plasmonic nanostructures[55]. The Poynting vector map reveals a complex multi-layer energy backflow structure, as shown in Fig. 4b. The energy flow vanishes at the positions of the electric and magnetic singularities and inherits their multi-layer matryoshka-like structure. Poynting vector vanishes at $z = 0$

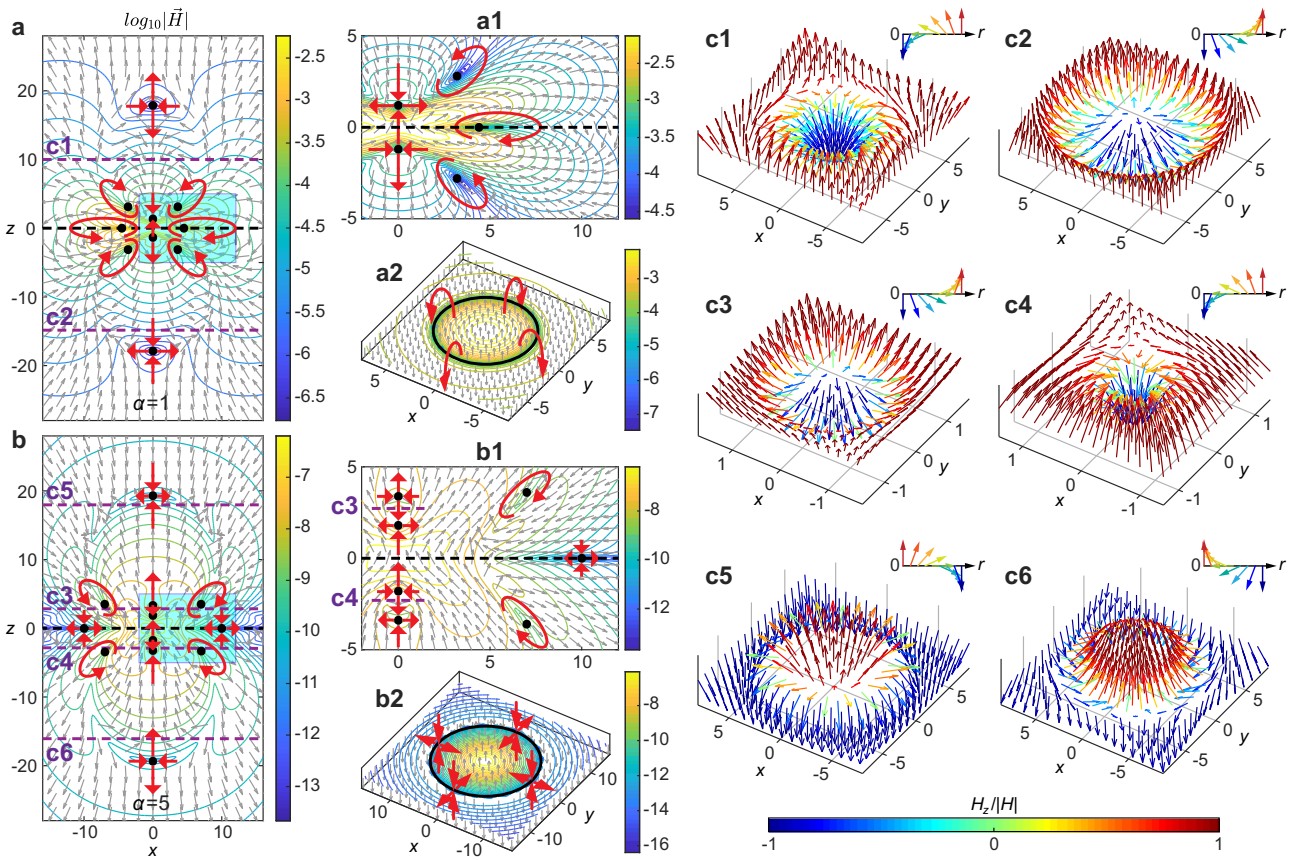

**Fig. 3 Magnetic fields of toroidal and supertoroidal light pulses. a, b** Isoline and arrow plots of the magnetic fields in the x–z plane for **a** the fundamental TLP and **b** the STLP of $\alpha = 5$, in logarithmic scale. Magnetic field singularities are marked by black dots with red arrows correspondingly marking the saddle or vortex style of the vector singularities. Panels **a1** and **b1** present the zoom-in plots corresponding to regions of blue boxes in (**a**) and (**b**), respectively. Panels **a2** and **b2** present transverse distributions of magnetic amplitude (in logarithmic scale) and normalized magnetic vectors at $z = 0$ planes, the positions marked by the black dashed lines in (**a**) and (**b**), respectively, where the magnetic fields vanish along the circular solid black lines with red arrows marking the styles of singularities (vortex for **a2** and saddle for **b2**). Skyrmionic structures in magnetic fields of toroidal and supertoroidal light pulses: **c** Various textures of Néel-type skyrmionic structure observed at various transverse planes (see dashed purple lines in **a** and **b**) for the fundamental TLP (**c1**–**c2**) and the STLP of $\alpha = 5$ (**c3**–**c6**), which are demonstrated by the arrows with color-labeled longitudinal component value of magnetic field. The up-right insert of each panel shows the basic texture of the skyrmionic structure. Unit for coordinates: $q_1$.

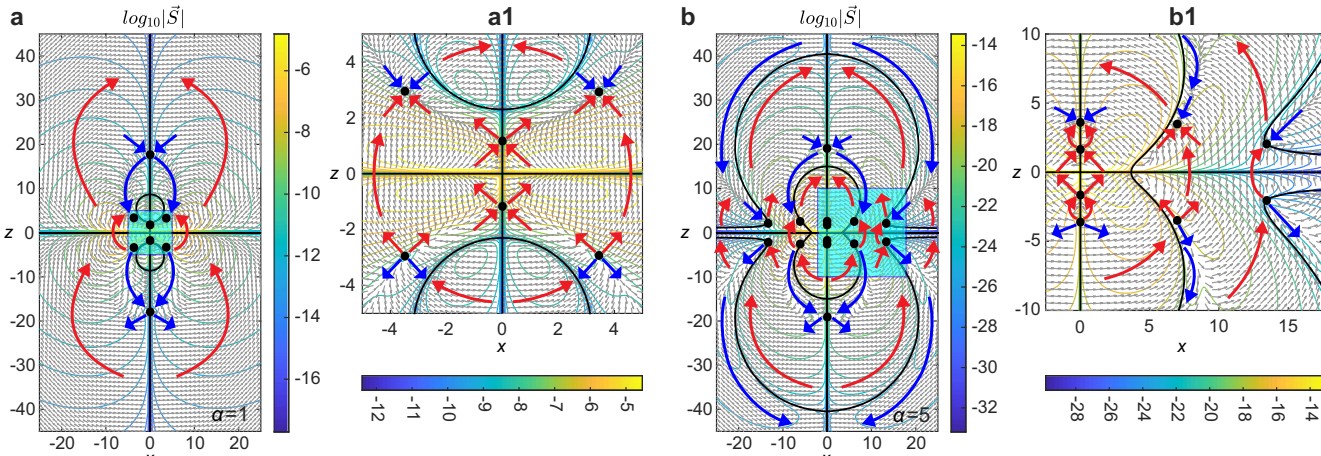

**Fig. 4 Poynting vector fields of toroidal and supertoroidal light pulses. a, b** Contour and arrow plots of the Poynting vector fields in the x–z plane, in logarithmic scale, for **a**) the fundamental TLP and **b** the STLP of $\alpha = 5$. Panels **a1** and **b1** present the zoom-in plots of the regions of blue boxes in (**a**) and (**b**), respectively. Poynting vector field zeros are marked by the black solid lines and black dots. Red and blue bold arrows highlight the regions of energy forward flow and backflow, respectively. Unit for coordinates: $q_1$.

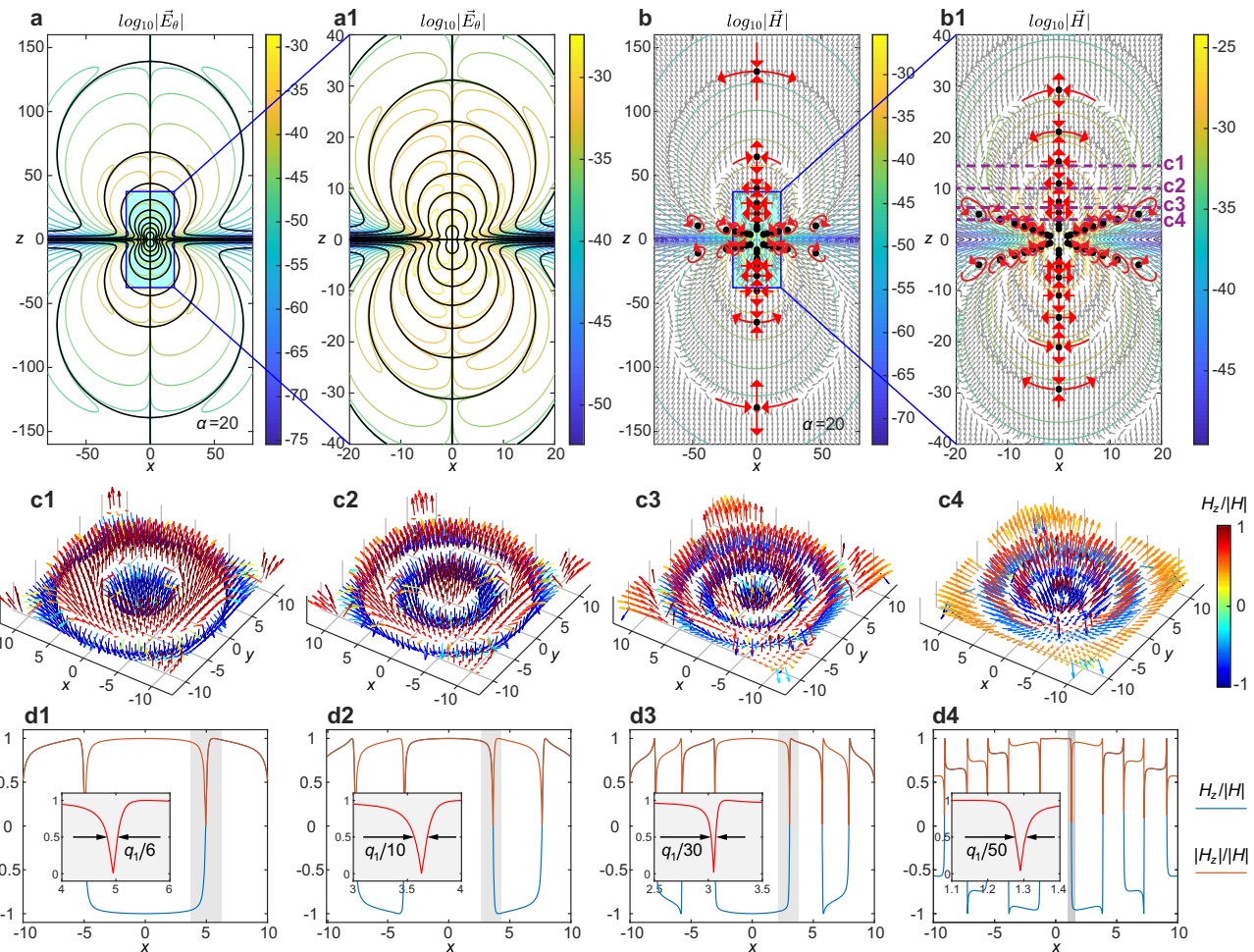

**Fig. 5 Fractal-like patterns in electromagnetic fields of supertoroidal pulses. a** The isoline plot of the electric field in the $x$–$z$ plane for the STLP of $\alpha = 20$, $\mathrm{Re}[E_{\theta}^{(\alpha=20)}(\mathbf{r}, t = 0)]$, in logarithmic scale. Electric field zeros are marked by the black solid lines and black dots. Panel **a1** presents the zoom-in plot of region highlighted by blue box in (**a**). **b** The isoline plot of magnetic field amplitude (in logarithmic scale) and arrow plot of normalized magnetic vectors in the $x$–$z$ plane for the STLP of $\alpha = 20$. Magnetic field singularities are marked by black dots with red arrows correspondingly marking the saddle or vortex style of the vector singularities. Panel **b1** presents the zoom-in plot of region highlighted in (**b**). Subwavelength features of skyrmionic structures: **c1–c4** The skyrmionic distributions of magnetic field at several transverse planes marked by dashed lines marked by "c1–c4" in (**b1**). **d1–d4** The distribution of normalized magnetic field and its absolute value versus $x$ for the skyrmionic structures in **c1–c4**. Insets illustrate the Subwavelength features at the regions highlighted by gray bands. Unit for coordinates: $q_1$.

plane, along the $z$-axis, and on the dual-layer matryoshka-like singular shells (marked by the black bold lines in Fig. 4b). Importantly, energy backflow occurs at areas of relatively low energy density, and, hence, STLP as a whole still propagates forward. For the temporal evolution of the energy flow of the pulse see Supplementary Videos 4 and 5.

**Fractal patterns hidden in electromagnetic fields.** As the order $\alpha$ of the pulse increases (see Supplementary Video 4), the topological features of the STLP appear to be organized in a hierarchical, fractal-like fashion. A characteristic case of the STLP of $\alpha = 20$ is presented in Fig. 5. For the electric field, the matryoshka-like singular shells involve an increasing number of layers as one examines the pulse at finer length scales, forming a self-similar pattern that seems infinitely repeated. For the magnetic field, the saddle and vortex points are distributed along the propagation axis and in two planes crossing the pulse center, respectively. The distribution of singularities becomes increasingly dense as one approaches the center of the pulse, resulting in a self-similar pattern. A similar pattern can be seen for the Poynting vector map (see Supplementary Videos 4 and 5).

**Deep-subwavelength features of skyrmionic structures.** The fractal-like pattern of vectorial magnetic field of a high-order STLPs results in skyrmionic configurations with features changing much faster than the effective wavelength $q_1$. Figures 5c1–c4 show the four skyrmionic structures of the high-order STLP ($\alpha = 20$) at the four transverse planes marked by the dashed lines in Fig. 5b1 at positions of $z/q_1 = 14, 10, 6, 3.5$, correspondingly. Here the observed skyrmionic structure is similar to the photonic skyrmion observed in ref. [20]. However, in contrast to the latter case, where skyrmionic structures were observed in the evanescent plasmonic field, here skyrmionic field configurations are observed in free-space propagating fields. Moreover, similarly to the "spin reversal" effect observed in deeply subwavelength scales in plasmonic skyrmionic fields[20], here we demonstrate "subwavelength" features in propagating skyrmionic fields at scales much smaller than the effective wavelength (cycle length) of the STLPs. The four skyrmionic structures we obtained Fig. 5c1–c4 have two different topologies with topological numbers of $(p, m, \gamma) = (1, 1, \pi)$, for c1 and c3, and $(-1, 1, 0)$, for c2 and c4. In addition, they exhibit an effect of "spin reversal", where the number reversals is given by $\bar{p} = \frac{1}{2\pi}[\beta(r)]_{r=0}^{r=\infty}$ ($\beta$ is defined as

the radially-variant angle, see details in Method), e.g., $\bar{p} = 1, 2, 3, 4$ for the skyrmionic structures in Fig. 5c1–c4, respectively. Each reversal corresponds to a sign change of $H_z$, which takes place over areas much smaller than the effective wavelength of the pulse ($q_1$). The full width at half maximum of these areas for the four skyrmionic structures is 1/6, 1/10, 1/30, 1/50 of the effective wavelength, respectively. Conclusively, the sign reversals become increasingly rapid in transverse planes closer to the pulse center ($z = 0$), see Fig. 5d1–d4. Similarly, increasing the value of $\alpha$ leads to increasingly sharper singularities. Notably, in contrast to the fundamental TLP, the skyrmionic configurations in STLP occur at areas of higher energy density, and thus we expect that they could be observed experimentally.

The topological structure of the STLP is directly related to the distribution of on-axis saddle points in its magnetic field. Indeed, the latter mark the intersection of the $E$-field singular shells with the $z$-axis, which in turn results in the emergence of different skyrmionic magnetic field patterns (see Fig. 5 and Videos 4 and 5). The number and position of on-axis magnetic field saddle points are defined by the supertoroidal parameter $\alpha$. This is illustrated in Fig. 6a, where we plot the number of on-axis $H$-field singularities as a function of alpha for a STLP with $q_2 = 20q_1$. The number of singularities is generally increasing with increasing $\alpha$ apart for values around $\alpha = 5.6$ (marked by blue dashed line in Fig. 6). Moreover, the number of singularities increases in a ladder-like fashion, where only specific values of alpha lead to additional singularities. The origin of this behavior can be traced to changes in the pulse structure as $\alpha$ increases (see Fig. 6b). For specific values of $\alpha$, additional singularities appear away from the pulse center ($z = 0$) and then move slowly towards it. On the other hand, the irregular behavior at $\alpha = 5.6$ is a result of two singularities disappearing (see blue dashed line in Fig. 6b). The topological structure of the STLP can be tuned also by the degree

of focusing, which is quantified by the ratio $q_2/q_1$. In particular, tightly focused pulses exhibit a more complex topological structure at finer scales as opposed to collimated pulses (see Supplementary Video 6).

## Discussion
STLPs exhibit complex and unique topological structure. The electric field exhibits a matryoshka-like configuration of singularity shells, which divide the STLP into "nested" regions with opposite azimuthal polarization. The magnetic field exhibits skyrmionic structures with multiple topological textures at various transverse planes of a single pulse, related to the distribution of multiple saddle and vortex singularities. The electric and magnetic fields can be exchanged respecting to the difference of TE and TM modes. The instantaneous Poynting vector field exhibits multiple singularities with regions of energy backflow. The singularities of the STLP appear to be hierarchically organized resulting in self-similar, fractal-like patterns for higher-order pulses.

The main challenges for the generation of supertoroidal pulses involve its toroidal topology, broad bandwidth (single-cycle duration), and complex spatially-dependent spectral structure (see Supplementary Note 5). We argue that supertoroidal pulses can be generated similarly to the generation of fundamental toroidal pulses[31,32], i.e., by conversion of ultrashort linearly polarized pulses in a two-stage process. This process shall involve the linear-to-radial polarization conversion of an ultrashort laser pulse, followed by the spatio-spectral modification of the pulse in a multi-layered gradient metasurface. We anticipate that the requirement for the single-cycle temporal profile will be possible to be met if we use attosecond laser pulses as input. Alternatively, in the THz range, single-cycle pulses can be routinely generated by optical rectification of femtosecond optical pulses.

In conclusion, to the best of our knowledge, STLPs are so far the only known example of free-space propagating skyrmions in electromagnetic field. Indeed, higher-order STLPs exhibit a range of different skyrmionic field configurations, which will be of interest in probing the topology of electromagnetic excitations in matter. Moreover, information can be encoded in the increasingly complex topological structure of the propagating pulses, which could be of interest for optical communications. Finally, the subwavelength features of the singular structures of the STLPs may lead to advanced approaches of super-resolution imaging and nanoscale metrology.

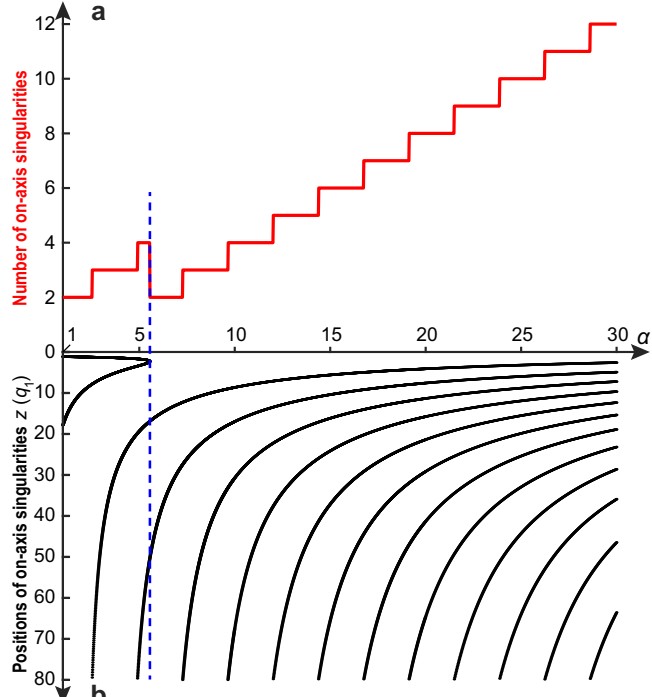

**Fig. 6 Evolution of on-axis singularity distribution versus supertoroidal order. a, b** The numbers (**a**) and positions (**b**) of on-axis saddle-singularities of the magnetic field of the STLP ($q_2 = 20q_1$) within the range of $z \in [0, 80q_1]$, versus $\alpha$. The blue dashed line marks where the number of singularities decreases.

## Methods
**Solving the supertoroidal pulses**. The first step is to solve the scalar generating function $f(\mathbf{r}, t)$ that satisfies the wave equation $(\nabla^2 - \frac{1}{c^2}\frac{\partial^2}{\partial t^2})f(\mathbf{r}, t) = 0$, where $\mathbf{r} = (r, \theta, z)$ are cylindrical coordinates, $t$ is time, $c = 1/\sqrt{\varepsilon_0\mu_0}$ is the speed of light, and the $\varepsilon_0$ and $\mu_0$ are the permittivity and permeability of medium. The exact solution of $f(\mathbf{r}, t)$ can be given by the modified power spectrum method as $f(\mathbf{r}, t) = f_0/[(q_1 + i\tau)(s + q_2)^\alpha]$, where $f_0$ is a normalizing constant, $s = r^2/(q_1 + i\tau) - i\sigma$, $\tau = z - ct$, $\sigma = z + ct$, $q_1$ and $q_2$ are parameters with dimensions of length and act as effective wavelength and Rayleigh range under the paraxial limit, while $\alpha$ is a real dimensionless parameter that must satisfy $\alpha \geq 1$ to ensure finite energy solutions. The next step is constructing the Hertz potential. For fulfilling the toroidal symmetric and azimuthally polarized structure, the Hertz potential should be constructed as $\mathbf{A}(\mathbf{r}, t) = \mu_0\nabla \times \hat{\mathbf{z}}f(\mathbf{r}, t)$. Then, the exact solutions of transverse electric (TE) and transverse magnetic (TM) modes are readily obtained by using Hertz potential. The electromagnetic fields for the TE solution can be derived by the potential as $\mathbf{E}(\mathbf{r}, t) = -\mu_0\frac{\partial}{\partial t}\nabla \times \mathbf{A}$ and $\mathbf{H}(\mathbf{r}, t) = \nabla \times (\nabla \times \mathbf{A})$[29,48], see Supplementary Notes 1–3 for more detailed derivations.

**Characterizing topology of skyrmions**. A skyrmion is a topologically stable 3D vector field confined within a 2D domain, noted as $\mathbf{n}(x, y)$, which can be represented as the vector distribution unwrapped from the vectors on a spiny sphere parametrized by longitude and latitude angles, $\alpha$ and $\beta$. The topological properties of a skyrmionic configuration can be characterized by the skyrmion number

defined by[24,51]:

$$s = \frac{1}{4\pi} \iint \mathbf{n} \cdot \left( \frac{\partial \mathbf{n}}{\partial x} \times \frac{\partial \mathbf{n}}{\partial y} \right) dx\, dy \qquad (7)$$

that is an integer counting how many times the vector $\mathbf{n}(x, y) = \mathbf{n}(r\cos\theta, r\sin\theta)$ wraps around the unit sphere. For mapping to the unit sphere, the vector can be given by $\mathbf{n} = (\cos\alpha(\theta)\sin\beta(r), \sin\alpha(\theta)\sin\beta(r), \cos\beta(r))$. Also, The skyrmion number can be separated into two integers:

$$\begin{aligned} s &= \frac{1}{4\pi} \int_0^\infty dr \int_0^{2\pi} d\varphi\, \frac{d\beta(r)}{dr}\frac{d\alpha(\theta)}{d\theta}\sin\beta(r) \\ &= \frac{1}{4\pi}[\cos\beta(r)]_{r=0}^{r=\infty}[\alpha(\theta)]_{\theta=0}^{\theta=2\pi} = p \cdot m \end{aligned} \qquad (8)$$

the polarity, $p = \frac{1}{2}[\cos\beta(r)]_{r=0}^{r=\infty}$, represents the direction of the vector field, down (up) at $r = 0$ and up (down) at $r \to \infty$ for $p = 1$ ($p = -1$). The vorticity number, $m = \frac{1}{2\pi}[\alpha(\theta)]_{\theta=0}^{\theta=2\pi}$, controls the distribution of the transverse field components. In the case of a helical distribution, an initial phase $\gamma$ should be added, $\alpha(\theta) = m\theta + \gamma$. For the $m = 1$ skyrmion, the cases of $\gamma = 0$ and $\gamma = \pi$ are classified as Néel-type, and the cases of $\gamma = \pm\pi/2$ are classified as Bloch-type. The case for $m = -1$ is classified as anti-skyrmion. See Supplementary Note 6 for some theoretically simulated results of skyrmions with various topological indices.

## Data availability

The data from this paper can be obtained from the University of Southampton ePrints research repository https://doi.org/10.5258/SOTON/D1901.

## Code availability

The code from this paper and details on the code used can be obtained from the University of Southampton ePrints research repository https://doi.org/10.5258/SOTON/D1901.

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

## Acknowledgements

The authors are grateful to Janne Ruostekoski for discussions and acknowledge the supports of the MOE Singapore (MOE2016-T3-1-006), the UKs Engineering and Physical Sciences Research Council (grant EP/M009122/1, Funder Id: https://doi.org/10.13039/501100000266), the European Research Council (advanced grant FLEET-786851, Funder Id: https://doi.org/10.13039/501100000781), and the Defense Advanced Research Projects Agency (DARPA) under the Nascent Light Matter Interactions program.

## Author contributions

Y.S. performed the theoretical modeling and numerical simulations and written the manuscript, Y.S., Y.H., N.P., and N.I.Z. contributed to data analysis and manuscript revision, and the project was supervised by N.I.Z.

## Competing interests

The authors declare no competing interests.
