## [Peer Review File · Nature Communications]

REVIEWER COMMENTS

Reviewer #1 (Remarks to the Author):

The topology of electromagnetic beams and pulses is at the focus of growing research efforts currently, in particular with respect to the possibility of studying and exploiting topological phenomena typically observed in condensed matter (e.g. skyrmions). Whereas a number of works have studied such phenomena in electromagnetic beams, not much is known about the topological properties of pulses. Thus, the presented results are important and will be of considerable interest to the community. Overall, the paper is well written, and I recommend it for publication. I also suggest the following optional improvement to the manuscript:

1. The authors may want to discuss in more detail the importance of their findings, in particular in the context of the recent stream of works (e.g. refs. [19,20,24]) on "skyrmion-like" electromagnetic field configurations.
2. The authors may want to qualitatively discuss the dependence of the topological structure on the q_1 & q_2 parameters.
3. The authors may want to suggest methods to generate the "supertoroidal pulses". Can they be generated in principle and what are the practical challenges?
4. The singularity structures presented in the Fig. 1 are not well explained. For instance, in the inset to Fig. 1a, what is annotated as line singularity appears to be a saddle point singularity.

Finally, there is a typo in the beginning of the Discussion section (line 4): "maagnetic".

Reviewer #2 (Remarks to the Author):

The authors introduce a complete theoretical work of a new family of structured light pulses, which possess many attractive topological properties, including the general toroidal topology, skyrmions, fractal patterns. Both the toroidal electrodynamics and optical skyrmions are increasingly hot topics in recent years, and I am pleased to see the presented modeling in this manuscript manage to find an intersection of these two subjects. Also, the pulse family refers to the structuring of few-cycle ultrafast pulses, thus it shows importance to open new research direction of ultrafast nonlinear optics. To me it is also the first demonstration of optical quasiparticle, i.e. the skyrmions in propagating freespace light pulses, bring new topological states of light. Therefore, this article has enough novelty that deserved to be published, after some revisions as suggested below.

(1) In my opinion, the work delivers two main novelties, the toroidal and the skyrmion. However, the backgrounds for both subjects are not included in the Introduction section. What is the importance of exploring new toroidal topology? What is the motivation of optical skyrmion? Besides, the authors have a paragraph introducing THz wave, which is confusing to me, since I don't get how THz motivate this work. Anyway, the article is well written except the Introduction section. Please reconsider the logic to give a more convincing introduction.

(2) There is a misleading citation in the last sentence of first paragraph, that reference [25] has nothing to do with spin-orbit optical forces, while it is a work on nucleon interaction. I suggest this citation being replaced or removed. I understand skyrmion is a very hot topic, but please be careful that many works are not in optics, and what discussed here is optical skyrmion.

(3) The "Results" part is very good, except a few expressions in the last subsection. For "fine-scale", how fine is the scale? Nanoscale or microscale? The author indeed explained this that it is sub- q_1 scale, and q_1 is effective wavelength, but one should avoid any ambiguous adjective in academic papers. Moreover, I found the sub-'wavelength' feature of skyrmion discussed here is extremely similar to a prior important work "Deep-subwavelength features of photonic skyrmions in a confined electromagnetic field with orbital angular momentum. Nat. Phys. 15, 650–654 (2019)." The similarity of two works should be discussed.

(4) The supplemental material includes a very clear and detailed derivation of supertoroidal pulses, but no basic theoretical framework on skyrmion. The Methods include a part introducing the skyrmion background,

but, as a reader, I am looking for a clearer theoretical background showing what is the ideal state of topological skyrmions. It helps if some simulation result can be provided so that one can compare the skyrmions in supertoroidal pulses with the ideal models. I suggest authors add some results in the supplemental material.

Reviewer #3 (Remarks to the Author):

The manuscript reports a family of few-cycle toroidal light pulses. The groundwork to this description was set in 1989, when Zielkowsky derived exact solution to Maxwell's equations in free space that combine properties of electromagnetic bullets and transient beams, including the defining equation of toroidal beams. Several properties of the simplest toroidal beams with azimuthal electric or magnetic fields were described soon after in 1996 by Hellwarth and Nouchi. The current manuscript extends this description to include higher order modes of toroidal light pulses, and describes phenomenologically several of their properties, including the spatial distribution of their vector fields, some considerations of the skyrmion character and energy backflow.

This is in principle an interesting development, and, without following the mathematical derivation in detail, I have no doubt that the work is correct.

What the manuscript lacks, in my opinion, is an explanation of why the described properties are important, and a motivation that goes beyond superficial hints at potential applications in information transfer or microscopy.

Where are the higher order modes superior?

What is the role of the defining parameter α , is there a fundamental difference between integer and fractal values of α , how does it impact on the skyrmion number or the energy backflow?

What are general considerations beyond describing features of the specific examples?

Are there experimental considerations for generating the more sophisticated modes?

I note that the "growing attention" that toroidal light pulses are apparently attracting is demonstrated with 10 papers sharing the final author with the current manuscript - a more balanced view would strengthen this point.

Finally, I suggest to replace the term "supertoroidal" with the more neutral "generalised toroidal light pulses".

Reviewer #1:

“The topology of electromagnetic beams and pulses is at the focus of growing research efforts currently, in particular with respect to the possibility of studying and exploiting topological phenomena typically observed in condensed matter (e.g. skyrmions). Whereas a number of works have studied such phenomena in electromagnetic beams, not much is known about the topological properties of pulses. Thus, the presented results are important and will be of considerable interest to the community. Overall, the paper is well written, and I recommend it for publication. I also suggest the following optional improvement to the manuscript:”

We thank the reviewer for their positive comments and recommendation for publication.

“1. The authors may want to discuss in more detail the importance of their findings, in particular in the context of the recent stream of works (e.g. refs. [19,20,24]) on "skyrmion-like" electromagnetic field configurations.”

We now include a comprehensive discussion of recent advances in the field of electromagnetic skyrmions in the Introduction, see page 1, left column, first paragraph.

“2. The authors may want to qualitatively discuss the dependence of the topological structure on the q_1 & q_2 parameters.”

In response to the reviewer's comment, we discuss the dependence of the topological structure on the q_1 and q_2 parameters in page 7, right column, end of first paragraph. We have also added two videos in the supplementary material to illustrate this dependence (see Supplementary Videos 3 & 6).

“3. The authors may want to suggest methods to generate the "supertoroidal pulses". Can they be generated in principle and what are the practical challenges? ”

The main challenges for the generation of supertoroidal pulses involve its toroidal topology, broad bandwidth (single-cycle duration), and complex spatially-dependent spectral structure (see Supplementary Information E). We argue that supertoroidal pulses can be generated similarly to the generation of fundamental toroidal pulses [30,31], i.e. by conversion of ultrashort linearly polarized pulses in a two-stage process. This process shall involve the linear-to-radial polarization conversion of an ultrashort laser pulse, followed by the spatio-spectral modification of the pulse in a multi-layered gradient metasurface. We anticipate that the requirement for the single-cycle temporal profile will be possible to be met if we use attosecond laser pulses as input. Alternatively, in the THz range, single-cycle pulses can be routinely generated by optical rectification of femtosecond optical pulses.

We discuss the generation of supertoroidal pulses in page 7, right column, 2nd paragraph of the Discussion section.

“4. The singularity structures presented in the Fig. 1 are not well explained. For instance, in the inset to Fig. 1a, what is annotated as line singularity appears to be a saddle point singularity.”

We thank the reviewer for pointing out this error. In the revised manuscript, we have corrected and expanded the description of singularities in Fig. 1: grey lines represent vortex-type singularities, solid circles indicate saddle-type singularities, and the colored vectors represent skyrmionic structures in selected transverse planes (see page 2, last paragraph before section “Electric field singularity”).

“Finally, there is a typo in the beginning of the Discussion section (line 4): "maagnetic".”

We have corrected this typo.

Reviewer #2:

“The authors introduce a complete theoretical work of a new family of structured light pulses, which possess many attractive topological properties, including the general toroidal topology, skyrmions, fractal patterns. Both the toroidal electrodynamics and optical skyrmions are increasingly hot topics in recent years, and I am pleased to see the presented modeling in this manuscript manage to find an intersection of these two subjects. Also, the pulse family refers to the structuring of few-cycle ultrafast pulses, thus it shows importance to open new research direction of ultrafast nonlinear optics. To me it is also the first demonstration of optical quasiparticle, i.e. the skyrmions in propagating freespace light pulses, bring new topological states of light. Therefore, this article has enough novelty that deserved to be published, after some revisions as suggested below. ”

We thank the reviewer for their supportive comments.

“ (1) In my opinion, the work delivers two main novelties, the toroidal and the skyrmion. However, the backgrounds for both subjects are not included in the Introduction section. What is the importance of exploring new toroidal topology? What is the motivation of optical skyrmion? Besides, the authors have a paragraph introducing THz wave, which is confusing to me, since I don't get how THz motivate this work. Anyway, the article is well written except the Introduction section. Please reconsider the logic to give a more convincing introduction. ”

We thank the reviewer for their suggestion. We have expanded our Introduction section to include a discussion of earlier work on the “toroidal” and “skyrmion” aspects of our work and have removed the paragraph on THz waves (see page 1, paragraphs 1 & 2).

“(2) There is a misleading citation in the last sentence of first paragraph, that reference [25] has nothing to do with spin-orbit optical forces, while it is a work on nucleon interaction. I suggest this citation being replaced or removed. I understand skyrmion is a very hot topic, but please be careful that many works are not in optics, and what discussed here is optical skyrmion. ”

We thank the reviewer for pointing out this error, which has been corrected in the revised manuscript.

“(3) The “Results” part is very good, except a few expressions in the last subsection. For “fine-scale”, how fine is the scale? Nanoscale or microscale? The author indeed explained this that it is sub- q_1 scale, and q_1 is effective wavelength, but one should avoid any ambiguous adjective in academic papers. Moreover, I found the sub-‘wavelength’ feature of skyrmion discussed here is extremely similar to a prior important work “Deep-subwavelength features of photonic skyrmions in a confined electromagnetic field with orbital angular momentum. Nat. Phys. 15, 650–654 (2019).” The similarity of two works should be discussed. ”

In response to the reviewer’s comment, we have replaced the phrase “fine-scale” with “subwavelength” and clearly state that the “wavelength” here refers to q_1 , the effective wavelength (cycle-length) of the pulse.

There is a dramatic difference between the subwavelength features presented in our work and that of Du et al., as our results involve free-space propagating waves rather than evanescent plasmonic fields. This is discussed in the Section “Subwavelength features of skyrmionic structures”.

“(4) The supplemental material includes a very clear and detailed derivation of supertoroidal pulses, but no basic theoretical framework on skyrmion. The Methods include a part introducing the skyrmion background, but, as a reader, I am looking for a clearer theoretical background showing what is the ideal state of topological skyrmions. It helps if some simulation result can be provided so that one can compare the skyrmions in supertoroidal pulses with the ideal models. I suggest authors add some results in the supplemental material. ”

We thank the reviewer for their suggestion. We have added a discussion of the theoretical framework of skyrmions in Supplementary Information F.

Reviewer #3:

“The manuscript reports a family of few-cycle toroidal light pulses. The groundwork to this description was set in 1989, when Zielkowsky derived exact solution to Maxwell's equations in free space that combine properties of electromagnetic bullets and transient beams, including the defining equation of toroidal beams. Several properties of the simplest toroidal beams with azimuthal electric or magnetic fields were described soon after in 1996 by Hellwarth and Nouchi. The current manuscript extends this description to include higher order modes of toroidal light pulses, and describes phenomenologically several of their properties, including the spatial distribution of their vector fields, some considerations of the skyrmion character and energy backflow. This is in principle an interesting development, and, without following the mathematical derivation in detail, I have no doubt that the work is correct. What the manuscript lacks, in my opinion, is an explanation of why the described properties are important, and a motivation that goes beyond superficial hints at potential applications in information transfer or microscopy.”

We thank the reviewer for their positive comments.

“Where are the higher order modes superior? ”

Supertoroidal pulses exhibit a number of advantages with respect to the fundamental Flying Doughnuts. Firstly, higher order toroidal pulses exhibit a range of different skyrmionic field configurations, which will be of interest for probing the topology of excitations in matter. Secondly, information can be encoded in the increasingly complex topological structure of the propagating pulses, which could be of interest for optical communications. Finally, the subwavelength features of the singular structures of the pulses may lead to new approaches to superresolution imaging and nanoscale metrology.

We include now this discussion in the final paragraph of the Discussion section, in page 7, right column.

“What is the role of the defining parameter alpha, is there a fundamental difference between integer and fractal values of alpha, how does it impact on the skyrmion number or the energy backflow? ”

The parameter α prescribes the degree of energy confinement in the pulse and is directly related to the finite energy requirement for the description of physical pulses. In particular, $\alpha < 1$ results in pulses of infinite energy, such as planar waves and cylindrical waves, while $\alpha \geq 1$ leads to finite-energy pulses. The parameter α is also related to the topology of the pulse. Indeed, given the few-cycle nature of the supertoroidal pulses, the increasing localization of energy for high values of α results in increasingly complex topological structures. Finally, in our study we found no evidence of a qualitative difference between integer and fractional values of α (see Supplementary Video 4).

We now include this discussion in page 1, first paragraph of the Results section.

“What are general considerations beyond describing features of the specific examples? ”

We added general considerations, especially in (1) Light-matter interactions; (2) Information transfer; (3) Imaging and metrology. See more details same as the response to the first comment.

“Are there experimental considerations for generating the more sophisticated modes?”

Please see our response to point 3 by reviewer #1.

“I note that the "growing attention" that toroidal light pulses are apparently attracting is demonstrated with 10 papers sharing the final author with the current manuscript - a more balanced view would strengthen this point. ”

In the revised manuscript, we provide a more comprehensive introduction on toroidal light with balanced citations (see added Refs. [34,35,38-42]).

“Finally, I suggest to replace the term "supertoroidal" with the more neutral "generalised toroidal light pulses". ”

We respectfully disagree with the reviewer. In the electrodynamics community, the term “supertoroidal” refers to generalizations of toroidal excitations (see for example Photonics 6, 43 (2019), PRA 98, 023858 (2018)). We now motivate the use of the term “supertoroidal” in the second paragraph of the Introduction section.

Reviewers' Comments:

Reviewer #1:

Remarks to the Author:

I am satisfied with author corrections and replies related to my comments. I recommend this paper for publication in Nature Communications.

Reviewer #2:

Remarks to the Author:

The authors have made a good revision work and well addressed all the concerns. Thus I recommend its publication in Nature Communications.

Reviewer #3:

None

Response to Reviewer #1:

“I am satisfied with author corrections and replies related to my comments. I recommend this paper for publication in Nature Communications.”

We thank the reviewer for their positive comments and recommendation for publication.

Response to Reviewer #2:

“The authors have made a good revision work and well addressed all the concerns. Thus I recommend its publication in Nature Communications.”

We thank the reviewer for their positive comments and recommendation for publication.